# Decarboxylation of *β*-boryl NHPI esters enables radical 1,2-boron shift for the assembly of versatile organoborons

Yu Guo[1], Xiaosha Wang[1], Chengbo Li[1], Jianke Su[1], Jian Xu[1] ✉ & Qiuling Song ◉[1,2,3] ✉

In recent years, numerous 1,2-R shift (R = aliphatic or aryl) based on tetra-coordinate boron species have been well investigated. In the contrary, the corresponding radical migrations, especially 1,2-boryl radical shift for the construction of organoborons is still in its infancy. Given the paucity and significance of such strategies in boron chemistry, it is urgent to develop other efficient and alternative synthetic protocols to enrich these underdeveloped radical 1,2-boron migrations, before their fundamental potential applications could be fully explored at will. Herein, we have demonstrated a visible-light-induced photoredox neutral decarboxylative radical cross-coupling reaction, which undergoes a radical 1,2-boron shift to give a translocated C-radical for further capture of versatile radical acceptors. The mild reaction conditions, good functional-group tolerance, and broad β-boryl NHPI esters scope as well as versatile radical acceptors make this protocol applicable in modification of bioactive molecules. It can be expected that this methodology will be a very useful tool and an alternative strategy for the construction of primary organoborons via a novel radical 1,2-boron shift mode.

Boron-containing compounds are important linchpins in organic synthesis[1–6] and also play an important role in material science and pharmaceuticals[7]. As the key intermediates, organoborons have been extensively employed in various cross-coupling reactions[8–12]. Among them, 1,2-metallate migration reactions and transmetallations of tetracoordinate boron species are the most prevalent and well-recognized reaction modes[11,13–18]. In fact, stereospecific 1,2-migrations of alkenylboronate complexes have been known for many decades. In 1967, Zweifel and co-workers reported the first 1,2-alkyl/aryl migrations of vinylboron "ate" complexes (e.g. alkenyl tetracoordinate boron species) induced by electrophilic halogenation[19]. In 2016, Morken and co-workers disclosed an enantioselective palladium-induced 1,2-alkyl/aryl migration of the same substrates[20–24]. More recently, Studer[25–31], Aggarwal[32–40], and Renaud[41] developed radical–polar

crossover reactions, in which 1,2-alkyl/aryl migrations of alkenyl tetracoordinate boron species are induced by alkyl radical additions (Fig. 1A). In recent years, radical borylations under transition-metal-free conditions, especially alkyl radical-involved borylations dramatically promote the expansion of organoborons. Although numerous 1,2-R shift (R = aliphatic or aryl) based on tetracoordinate boron species have been well investigated, the corresponding radical migrations, especially 1,2-boryl radical shifts are underdeveloped and elusive. There are only very few cases describing such migrations[42]. For instance, in 1999, Batey and Smil disclosed that boron-tethered radical cyclizations could undergo a 1,2-boryl radical shift (Fig. 1B, eq. a)[43], however, no significant progress has been achieved until 2019, in that year Aggarwal and coworkers reported a photoredox catalysis-promoted activation of 1,2-bis-boronic esters to lead to primary

[1]Institute of Next Generation Matter Transformation, College of Material Sciences Engineering, Huaqiao University, 361021 Xiamen, Fujian, P. R. China. [2]Key Laboratory of Molecule Synthesis and Function Discovery, Fujian Province University, College of Chemistry at Fuzhou University, 350108 Fuzhou, P. R. China. [3]School of Chemistry and Chemical Engineering, Henan Normal University, 453007 Xinxiang, Henan, P. R. China. ✉e-mail: jianx@hqu.edu.cn; qsong@hqu.edu.cn

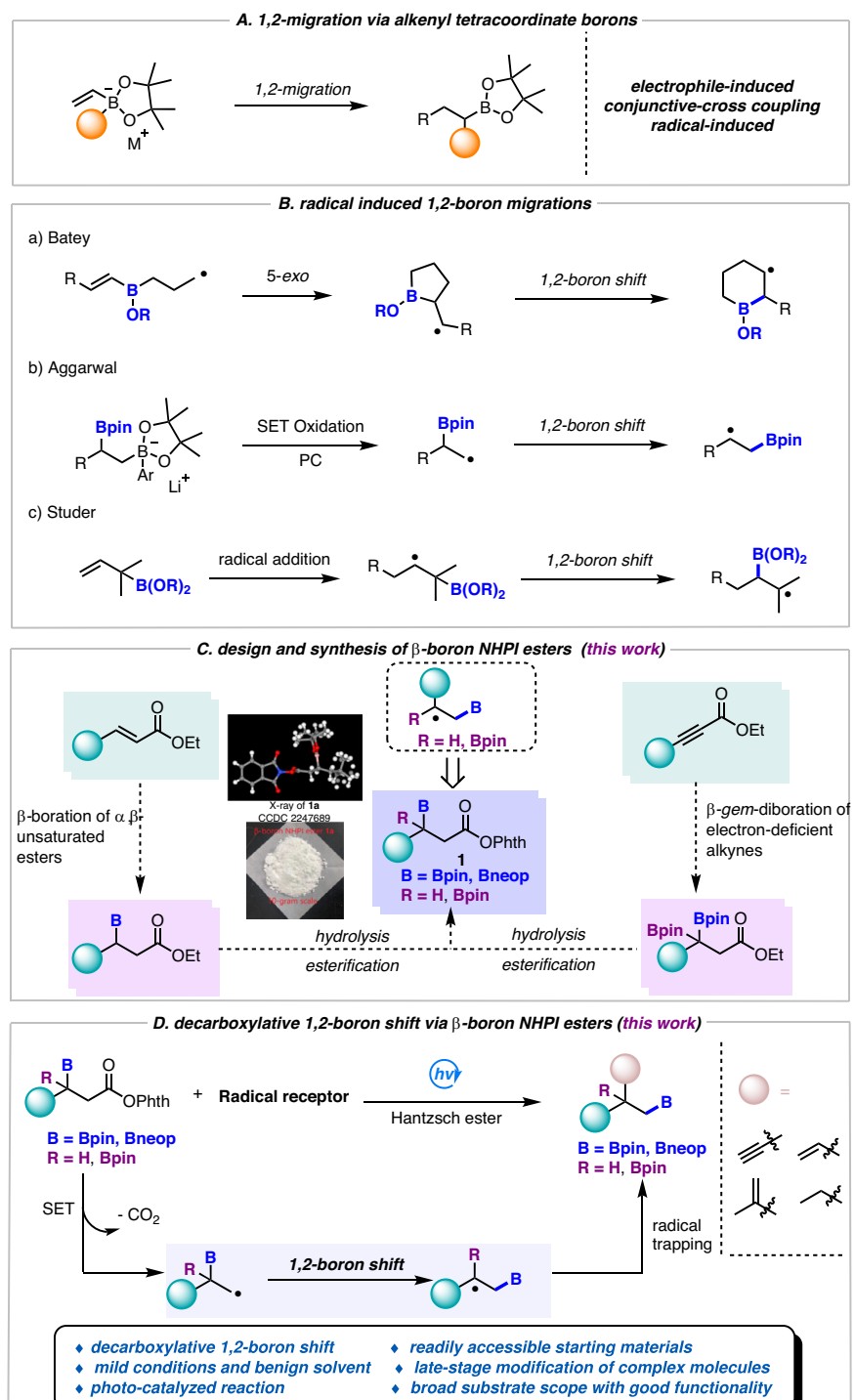

**Fig. 1 | Known 1,2-migration strategies of alkenyl tetracoordinate boron species, radical induced 1,2-boron migration, and our design.** A 1,2-Migration via alkenyl tetracoordinate borons. **B** Radical-induced 1,2-boron migrations. **C** Design and synthesis of β-boryl NHPI esters (this work). **D** Decarboxylative 1,2-boron shift of β-boryl NHPI esters (this work).

β-boryl radicals, which further undergo rapid 1,2-boron shift to form thermodynamically favored secondary radicals, finally affording significant functionalized boronic esters (Fig. 1B, eq. b)[44–46]. Later on, alkynyl triflones as CF$_3$-radical precursors as well as the subsequent alkynylation reagents are employed for the trifunctionalization of allylic boronates, which represents a new 1,2-boryl radical migration strategy was disclosed by Studer and co-workers (Fig. 1B, eq. c)[47,48]. Despite these achievements, efficient 1,2-boryl radical migration for the construction of organoborons is still in its infancy. Given the

paucity and significance of such strategies in boron chemistry, it is urgent to develop other efficient and alternative synthetic protocols to enrich these underexplored radical 1,2-boron migrations from readily accessible starting materials, before their fundamental potential applications can be fully explored at will.

*N*-Hydroxyphthalimide (NHPI) esters, which could be readily accessible from the corresponding carboxylic acids, have emerged as efficient alkyl radical precursors to be coupled with radical acceptors, nucleophiles (such as amines, alcohols, borons, etc) or

**Table 1 | Optimization of the reaction conditions[a]**

| Entry | PC | HE (x equiv) | DIPEA (y equiv) | Solvent | Yield (%)[b] |
|---|---|---|---|---|---|
| 1 | Ir(ppy)$_3$ | 1.5 | 2 | DCM | 52 |
| 2 | [Ir(dtbppy)(ppy)$_2$]PF$_6$ | 1.5 | 2 | DCM | 48 |
| 3 | [Ir{dFCF$_3$ppy}$_2$(bpy)]PF$_6$ | 1.5 | 2 | DCM | 40 |
| 4 | 4CzIPN | 1.5 | 2 | DCM | 41 |
| 5 | Ru(bpy)$_3$Cl$_2$ | 1.5 | 2 | DCM | 33 |
| 6 | Ru(bpy)$_3$Cl$_2$·6H$_2$O | 1.5 | 2 | DCM | 26 |
| 7 | Ru(bpy)$_3$(PF$_6$)$_2$ | 1.5 | 2 | DCM | 24 |
| 8 | Ir(ppy)$_3$ | 1.5 | 2 | THF | 45 |
| 9 | Ir(ppy)$_3$ | 1.5 | 2 | CH$_3$CN | 39 |
| **10** | **Ir(ppy)$_3$** | **1.5** | **2** | **DCE** | **74** |
| 11 | Ir(ppy)$_3$ | 1.5 | 2 | toluene | 29 |
| 12 | Ir(ppy)$_3$ | 1 | 2 | DCE | 61 |
| 13 | Ir(ppy)$_3$ | 2 | 2 | DCE | 56 |
| 14 | Ir(ppy)$_3$ | 1.5 | 1 | DCE | 48 |
| 15 | Ir(ppy)$_3$ | 1.5 | 3 | DCE | 52 |
| 16 | – | 1.5 | 2 | DCE | Trace |
| 17 | Ir(ppy)$_3$ | – | 2 | DCE | Trace |
| 18 | Ir(ppy)$_3$ | 1.5 | – | DCE | 38 |
| 19[c] | Ir(ppy)$_3$ | 1.5 | 2 | DCE | Trace |

HE (diethyl 1,4-dihydro-2,6-dimethyl-3,5-pyridinedicarboxylate), DIPEA (N, N-diisopropylethylamine), DCM (dichloromethane), DCE (1,2-dichloroethane), THF (tetrahydrofuran)
[a]Reaction conditions: **1a** (0.2 mmol), **2a** (1.5 equiv, 0.3 mmol), photocatalyst (1 mol%, 0.002 mmol), HE (1.5 equiv, 0.3 mmol), DIPEA (2 equiv, 0.4 mmol), solvent (2 mL) at room temperature, 40 W blue LEDs, 12 h in argon.
[b]Isolated yield.
[c]Without light.

electrophiles[49–64]. Therefore, we decided to use boron-containing NHPI esters named β-boryl NHPI esters as radical precursors, presumably, decarboxylation of such boron compounds under the action of light would produce β-boryl alkyl radicals, which could be used for further investigations of the elusive radical 1,2-boron migration reactions.

According to retrosynthetic analysis, β-boryl NHPI esters could be readily accessible from β-boryl esters, which are the result of Cu-catalyzed β-boration of various α,β-unsaturated carbonyl compounds[65]. After extensive efforts, we found that β-boryl NHPI ester **1a** could be practically obtained on a 10-gram scale with a 66% overall yield. The structure of **1a** is confirmed by X-ray crystal structure analysis (Fig. 1C, left). Furthermore, we were intrigued as to whether our approach could be extended to the selective functionalization of β-gem-diboryl NHPI esters. In this proposed strategy, a radical 1,2-boron migration would enable a thermodynamically favored boron-bearing tertiary alkyl radical, thus resulting in 1,2-diboron compounds. Remarkably, the β-boryl NHPI esters and β-gem-diboryl NHPI esters are white solids and are stable in air for months. Thus, the advantages of such compounds make them practical and accessible to synthetic and medicinal researchers. In the presence of photo-catalysis, these β-boryl NHPI esters could undergo facile single-electron transfer followed by rapid decarboxylative fragmentation to lead to a β-boryl alkyl radical, which then might engage in a radical 1,2-boron shift. The migration will be steered by thermodynamic effects with the increased stability of the rearranged C-radical. Trapping of the translocated rearranged radical

via cross-coupling reactions could render versatile mono primary boronic esters with β-boryl NHPI esters (Fig. 1D). Of note, besides various secondary β-boryl NHPI esters, β-gem-diboryl NHPI esters are well tolerated under this transformation as well, thus rendering various aliphatic 1,2-diborons. The mild reaction conditions, good functional-group tolerance, and broad β-boryl and β-gem-diboryl NHPI esters scope as well as versatile radical acceptors make this protocol applicable in the modification of bioactive molecules.

## Results
To validate our hypothesis, we initiated an optimization study of a model reaction between 1,3-dioxoisoindolin-2-yl 4,4-dimethyl-3-(4,4,5,5-tetramethyl-1,3,2-dioxaborolan-2-yl)pentanoate (**1a**) and ((phenylethynyl)sulfonyl)benzene (**2a**) in the presence of a photo-catalyst and a base under a blue LED lamp (Table 1). The desired cross-coupling product **3a** was obtained in 52% isolated yield when using Ir(ppy)$_3$ as a photocatalyst, diisopropylethylamine (DIPEA)/Hantzsch ester (HE) as the reductants, DCM as a solvent, and under the irradiation of a 40 W blue LED lamp (entry 1, Table 1). A screening of photocatalysts, such as [Ir(dtbppy)(ppy)$_2$]PF$_6$, [Ir{dFCF$_3$ppy}$_2$(bpy)]PF$_6$, 4CzIPN, and so on, demonstrated that Ir(ppy)$_3$ was the most effective catalyst, yielding **3a** in 52% yield (entries 2–7, Table 1. Also see Supplementary Information (SI) for details). Solvent examinations showed that DCE was the optimal one to deliver the target product **3a** in 74% yield (entries 8–11, Table 1). Further assessment on the amount of

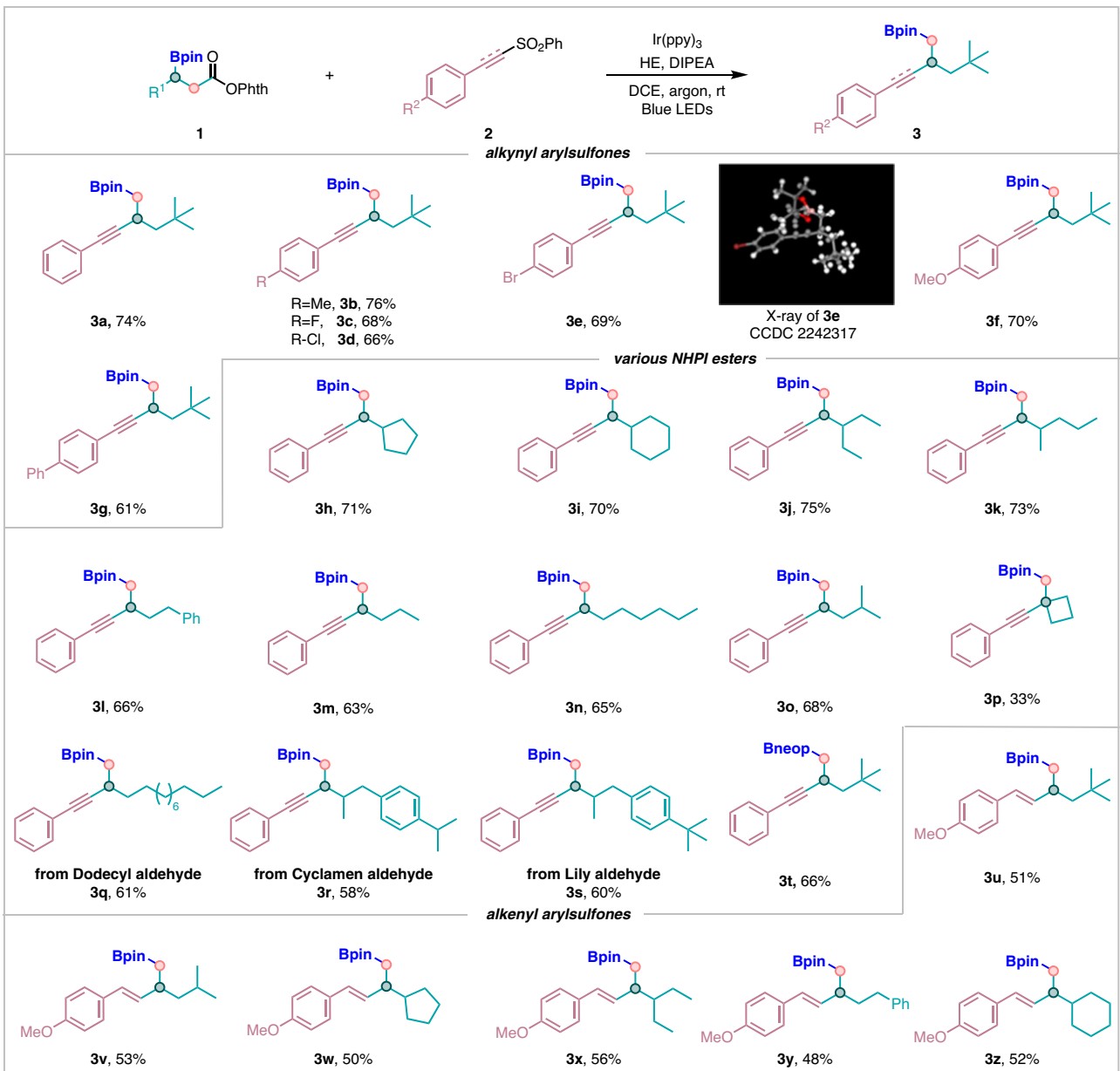

**Fig. 2 | Substrate scope and functional group compatibility of the decarboxylative alkynylation and alkenylation.** Reaction conditions: **1** (0.2 mmol), **2** (1.5 equiv, 0.3 mmol), Ir(ppy)₃ (1 mol%, 0.002 mmol), HE (1.5 equiv, 0.3 mmol), DIPEA (2 equiv, 0.4 mmol), DCE (2 mL) at room temperature, 40 W blue LEDs, 12 h in argon.

reductants indicated that 1.5 equiv. HE and 2 equiv. DIPEA were the best combination (entries 12–15, Table 1). Finally, control experiments demonstrated that the visible light, photocatalyst and HE were all necessary for this transformation, as no desired product **3a** was obtained in the absence of any of the above reaction promoters (entries 16–19, Table 1).

**Substrate scopes**

With the optimum conditions in hand (entry 10, Table 1), the scope of this decarboxylative alkynylation of β-boryl NHPI esters was explored and the results were shown in Fig. 2. This protocol was applicable to a wide range of alkynyl arylsulfones with either electron-donating or electron-withdrawing substituents on the benzene rings, affording the corresponding products **3a–3g** in moderate to good yields (61–76%). Next, various secondary β-boryl NHPI esters were prepared and subjected to the standard conditions, to our delight, they could react smoothly with alkynyl arylsulfone **2** and delivered the target products

**3h–3s** in decent yields. For instance, in terms of β-boryl NHPI esters bearing a secondary carbon on α-position, both cyclic (including five-membered and six-membered cyclic rings) and acyclic alkyl groups were compatible and converted into corresponding alkynylation products in moderate yields (**3h–3k**). For β-boryl NHPI esters bearing a primary carbon on α-position, phenylethyl (**1l**) and isopropyl ethyl (**1o**) were also suitable candidates for this transformation and the corresponding products **3l** and **3o** were obtained in 66% and 68% yields, respectively. To understand the impact of the stability of radicals on the reactivity and selectivity, substrates (**1m, 1n**) containing methylene tethers with different lengths were exposed to the reaction as well, it was found that the length of the chain does not affect the formation of the desired products. Moreover, for the tertiary alkyl group in β-boryl NHPI esters (**1p**), the corresponding target product **3p** could also be obtained successfully albeit in a lower yield. To showcase the applicability and practicality of this method, we also employed the late-stage modifications of some bioactive compounds or drug molecules. For

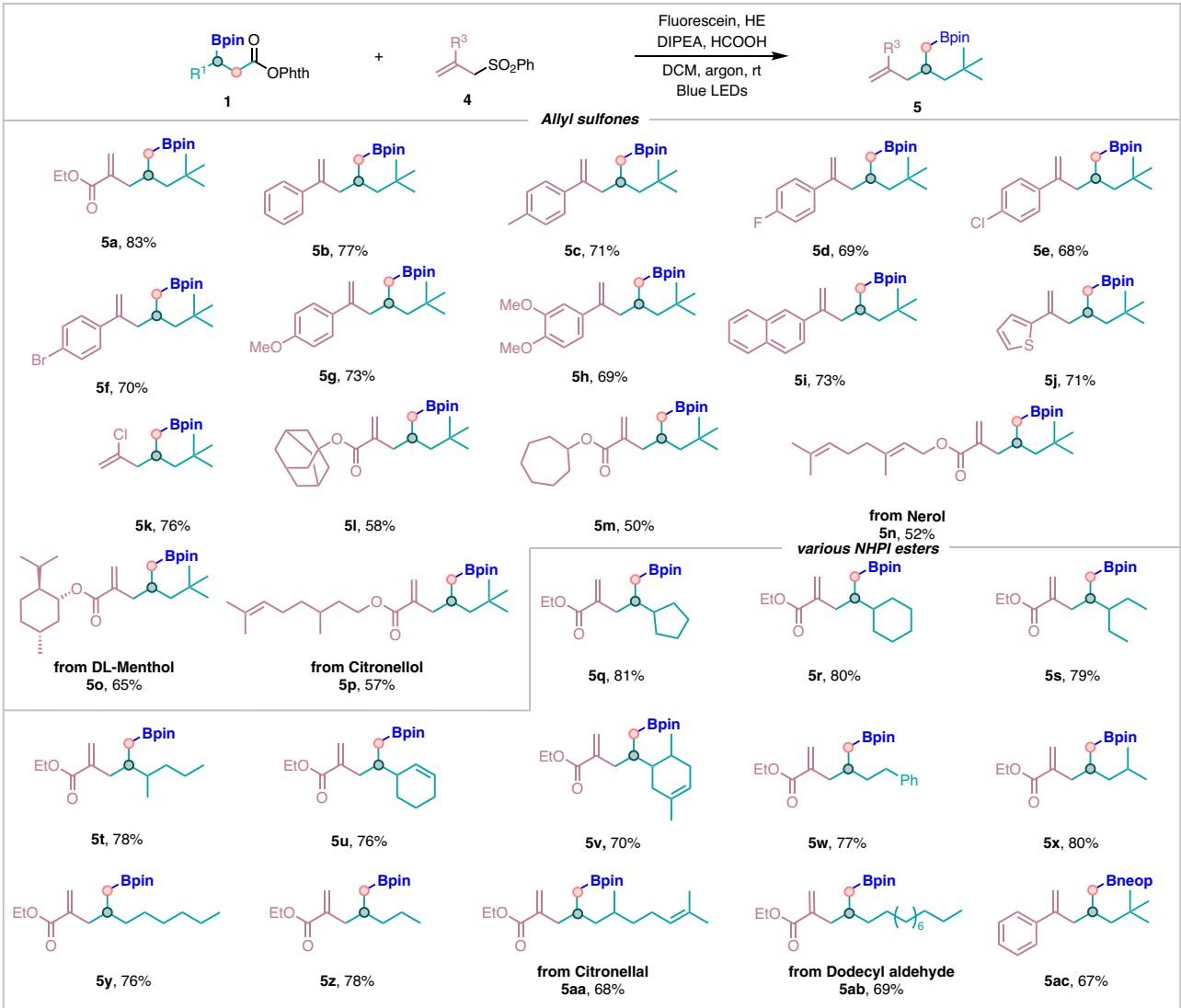

**Fig. 3 | Substrate scope and functional group compatibility of the decarboxylative allylation.** Reaction conditions: **1** (0.2 mmol), **4** (1.5 equiv, 0.3 mmol), Fluoresein (1 mol%, 0.002 mmol), HE (2 equiv, 0.4 mmol), DIPEA (1 equiv, 0.2 mmol), HCOOH (1 equiv, 0.2 mmol), DCM (2 mL) at room temperature, 40 W blue LEDs, 12 h in argon.

example, a series of natural products (dodecyl aldehyde, Cyclamen aldehyde, Lily aldehyde) were also successfully introduced into the $\beta$-boryl NHPI esters, and they all reacted well with alkynyl arylsulfones and produced the corresponding products (**3q**, **3r** and **3s**) without loss of efficiency. Moreover, we have examined $\beta$-boryl NHPI esters (using $B_2neop_2$ as the boron source for 1,2-boron shift). The substrate was also compatible with our reaction system to deliver the desired product **3t** in decent yield. After establishing the approach for the construction of C$sp^3$–C$sp$ bonds, we were especially interested in forging C$sp^3$–C$sp^2$ bonds by using vinyl sulfones as coupling partners. As expected, these substrates were equipotent to afford the corresponding coupling products (**3u**–**3z**) with C$sp^2$–C$sp^3$ bond formation in moderate yields under the standard conditions just like with alkynyl arylsulfones as substrates.

Encouraged by the aforementioned results, we subsequently evaluated the scope of allyl sulfones (Fig. 3). It turned out that allyl sulfones were suitable radical acceptors as well, and these substrates demonstrated good reactivities, and the corresponding coupling products were rendered in moderate to excellent yields (**5a**–**5ab**) under reoptimized conditions as follows: using fluorescein as photocatalyst, HE and DIPEA as the reductants, DCE as solvent, and under the

irradiation of a 40 W blue LED lamp base at rt for 12 h under argon atmosphere (see SI for details). The results were summarized in Fig. 3. We found that a broad range of allyl sulfones were compatible substrates for this decarboxylative allylation of $\beta$-boryl NHPI esters, and the corresponding desired products were procured with yields ranging from 50% to 83% (**5a**–**5p**). Both electron-donating and electron-withdrawing substituents on the *para* and *meta* positions of the benzene rings (**5b**–**5h**) worked well in our catalytic system to afford the corresponding allylation products in moderate to excellent yields. This catalytic system was also amenable to naphthalene and heterocycles, and the corresponding products **5i**–**5j** were isolated in good yields. In addition to aryl-substituted substrates, other allyl sulfones, such as halogenated and ester-substituted allyl sulfones, have also been confirmed to participate in the reactions well to obtain the corresponding homoallylic boronates (**5k**–**5p**). Allyl sulfones derived from structurally complicated natural products such as nerol, DL-menthol and citronellol, all reacted well and produced the corresponding products in synthetically useful yields (**5n**–**5p**). It is indicated that various substituted $\beta$-boryl NHPI esters and ethyl 2-((phenylsulfonyl)methyl) acrylate were also well-suited for this transformation (**5q**–**5ab**). For boron sources containing other substituents (using $B_2neop_2$ as the

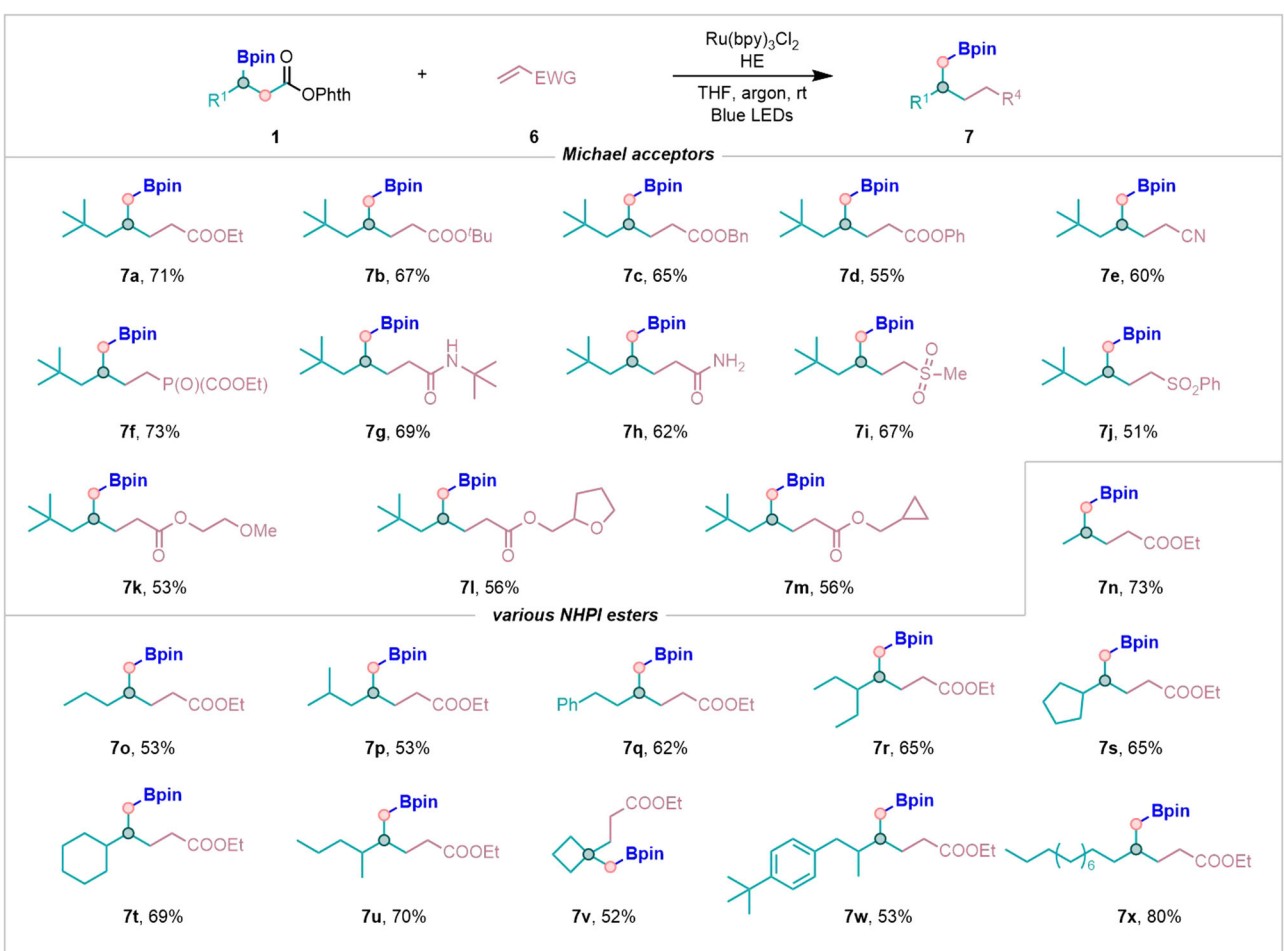

**Fig. 4 | Substrate scope and functional group compatibility of the decarboxylative alkylation.** Reaction conditions: **1** (0.2 mmol), **6** (1.5 equiv, 0.3 mmol), Ru(bpy)₃Cl₂ (1 mol%, 0.002 mmol), HE (1.5 equiv, 0.3 mmol), THF (2 mL) at room temperature, 40 W blue LEDs, 12 h in argon.

boron source for 1,2-boron shift), it was also compatible with this reaction system to render the target product **5ac** in good yield.

To further extend the scope of the present method, our interest subsequently shifted to investigation of other radical acceptors (Fig. 4). Radical conjugate addition to a range of electron-deficient alkenes afforded the desired products (**7a–7m**) in good to excellent yields. In addition to ethyl acrylates, a series of alkyls, such as *tert*-butyl, benzyl, phenyl acrylates and other substituent groups were viable substrates to give the corresponding products (**7a–7d**, **7k–7m**) as well. Furthermore, this photoredox system could enable decarboxylative alkylation of the *β*-boryl NHPI esters with other electron-deficient alkenes such as acrylonitrile, vinyl phosphonates, vinyl amides, vinyl sulfones with good efficiency (**7e–7j**). For *β*-boryl NHPI esters, various substrates could furnish the desired products (**7n–7x**) smoothly.

Interestingly, with our current strategy, when *β-gem*-diboryl NHPI esters were employed as the substrates, the reaction worked perfectly to lead to the corresponding new 1,2-diboron products. As summarized in Fig. 5, allyl sulfones regardless of aromatic ring or chlorine atom substituents all proved to be suitable substrates, leading to the desired products in good yields (**8a–8l**). Of note, alkynyl phenylsulfones were well tolerated to afford the desired products in moderate yields (**8m–8o**).

### Downstream applications and transformations
To demonstrate the synthetic utility of this coupling protocol, several scale-ups and transformations were performed, shown in Fig. 6. First of all, the scalabilities were performed by gram-scale reactions on **3a**, **5a**, and **7a**, without significant erosion of efficiency (Fig. 6A). In order to

showcase practical values of this strategy, then a series of transformations on **3a** were conducted. For example, firstly, the boron moiety could be oxidized into hydroxyl groups to lead to **9** in 85% yield (Fig. 6B, eq. a). Then, the boron moiety could also be retained through conversion to the corresponding potassium trifluoroborate salt **10**, which can be conveniently isolated (Fig. 6B, eq. b). Next, carbon homologations were achieved through the incorporation of an alkene group **11** at the C-B terminus with Grignard reagents (Fig. 6B, eq. c). The **3a** was smoothly converted to aryl groups via Suzuki–Miyaura couplings under the identified conditions, rendering the product **12** in 68% yield and the product **13** in 53% yield (Fig. 6B, eq. d). The introduction of (hetero)aryl motifs was within reach by treatment with organolithium reagents (**14** and **15**, Fig. 6B, eq. e and eq. f). And the alkyne **4a** could undergo complete hydrogenation with Pd/C catalyst, providing **16** in 90% yield (Fig. 6B, eq. g), which features a saturated aliphatic chain.

### Mechanism investigations
To gain more insights into this transformation, we carried out some control experiments and mechanistic studies. The reaction was completely suppressed in the presence of radical scavengers such as TEMPO, BHT, or 1,1-diphenylethylene, and the radical trapping products **17–19** could be detected by HRMS (Fig. 7a). The above results revealed that this process might proceed via a radical pathway. We subsequently turned our attention to the radical-clock experiments. Compound **20** undertook decarboxylation to produce a radical, which participated in our transformation through the way of intramolecular cyclization, thus confirming the radical process (Fig. 7b).

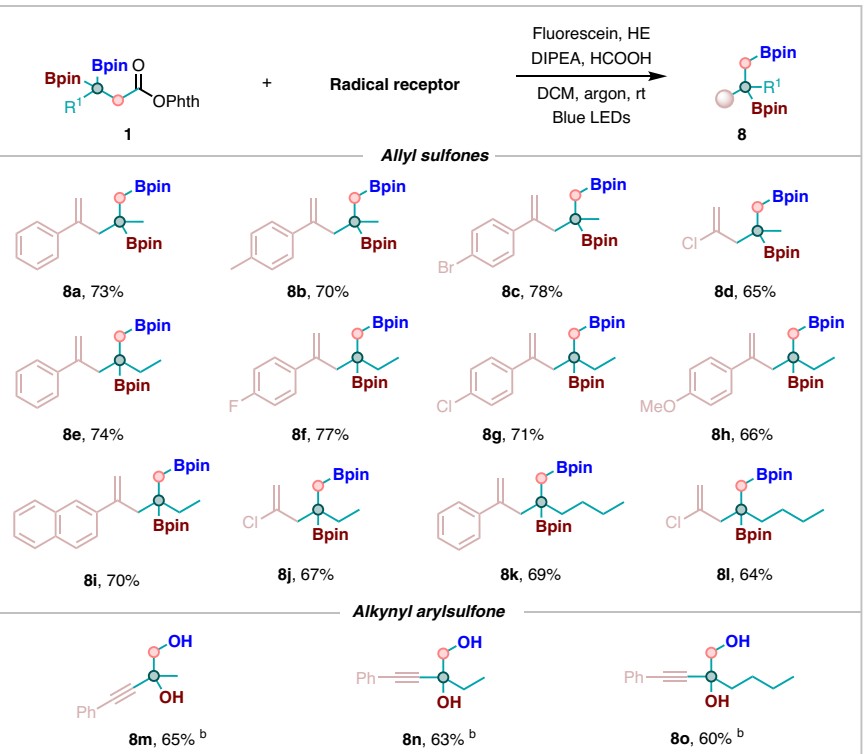

**Fig. 5 | Substrate scope and functional group compatibility of β-gem-diboryl NHPI esters.** Reaction conditions: [a] **1** (0.2 mmol), **4** (1.5 equiv, 0.3 mmol), Fluoresein (1 mol%, 0.002 mmol), HE (2 equiv, 0.4 mmol), DIPEA (1 equiv, 0.2 mmol), HCOOH (1 equiv, 0.2 mmol), DCM (2 mL) at room temperature, 40 W blue LEDs, 12 h in argon. [b] **1** (0.2 mmol), **2** (1.5 equiv, 0.3 mmol), Ir(ppy)₃ (1 mol%, 0.002 mmol), HE (1.5 equiv, 0.3 mmol), DIPEA (2 equiv, 0.4 mmol), DCE (2 mL) at room temperature, 40 W blue LEDs, 12 h in argon.

Based on the above control experiments, a possible reaction mechanism is proposed as outlined in Fig. 8. Photocatalyst Ir(ppy)₃ is excited to photoexcited Ir(III)* via single-electron transfer under visible light, the latter is then reduced by Hantzsch ester to Ir(II). Single-electron transfer from visible light-excited photocatalyst Ir(III) to the N-hydroxyphthalimide **1** generates a radical anion **A**, which then undergoes homolytic cleavage of the N–O bond to deliver alkyl radical **B** by decarboxylation and the release of phthalimide anion. The alkyl radical **B** thus engages in a 1,2-boron shift to form **C**. After α-addition to the alkynyl sulfone **2** and the sulfonyl radical elimination, the alkyne product **3** is obtained. With the proton transfer from the Hantzsch ester radical cation, β-boryl alkyl radical then adds either to the allyl sulfone **4** to obtain the allylation adduct **5** after desulfonation or to the Michael acceptor to obtain the Michael addition adduct **7**.

## Discussion

In conclusion, we have demonstrated a visible-light-induced photoredox neutral radical decarboxylative cross-coupling reaction, which undergoes a radical 1,2-boron shift to give a translocated C-radical for further capture of versatile radical acceptors. The method features mild conditions and simple operations, avoids the use of highly active organometallic reagents, and enriches the diversity of substrates. Given the ready accessibility of the starting materials, the operational simplicity, and the valuable products, it can be expected that this methodology will be a very useful tool and an alternative strategy for the construction of organoborons via a novel radical 1,2-boron shift mode. Furthermore, β-gem-diboryl NHPI esters undergo a 1,2-boron shift to give 1,2-dibrons or 1,2-diols after oxidation.

## Methods
### General procedure for synthesis of primary boronates from alkynyl arylsulfones or vinyl sulfones

A mixture of **1** (0.2 mmol), **2** (0.3 mmol), Ir(ppy)₃ (1% mmol), and HE (0.3 mmol) were charged into a Schleck tube, then the air was removed, argon was filled of Schleck tube and DIPEA (0.2 mmol), DCE (2 mL) is added the mixture. The mixture was stirred under irradiation from 40 W Blue LEDs. After the solvent was removed under reduced pressure, the residue was purified by silica gel chromatography using PE/EA (50:1) to afford the corresponding product.

### General procedure for synthesis of primary boronates from allyl sulfones

A mixture of **1** (0.2 mmol), fluorescein (1% mmol), and HE (0.4 mmol) was charged into a Schleck tube, then the air was removed, argon was filled in a Schleck tube and **4** (0.3 mmol), DIPEA (0.2 mmol), HCOOH (0.2 mmol), DCM (2 mL) is added the mixture. The mixture was stirred under irradiation from 40 W Blue LEDs. After the solvent was removed under reduced pressure, the residue was purified by silica gel chromatography using PE/EA (50:1) to afford the corresponding product.

### General procedure for synthesis of primary boronates from electron-deficient alkenes

A mixture of **1** (0.2 mmol), Ru(bpy)₃Cl₂ (1% mmol), and HE (0.3 mmol) was charged into a Schleck tube, then the air was removed, argon was filled in a Schleck tube and **6** (0.3 mmol), THF (2 mL) is added the mixture. The mixture was stirred under irradiation from 40 W Blue LEDs. After the solvent was removed under reduced pressure, the residue was purified by silica gel chromatography using PE/EA (30:1) to afford the corresponding product.

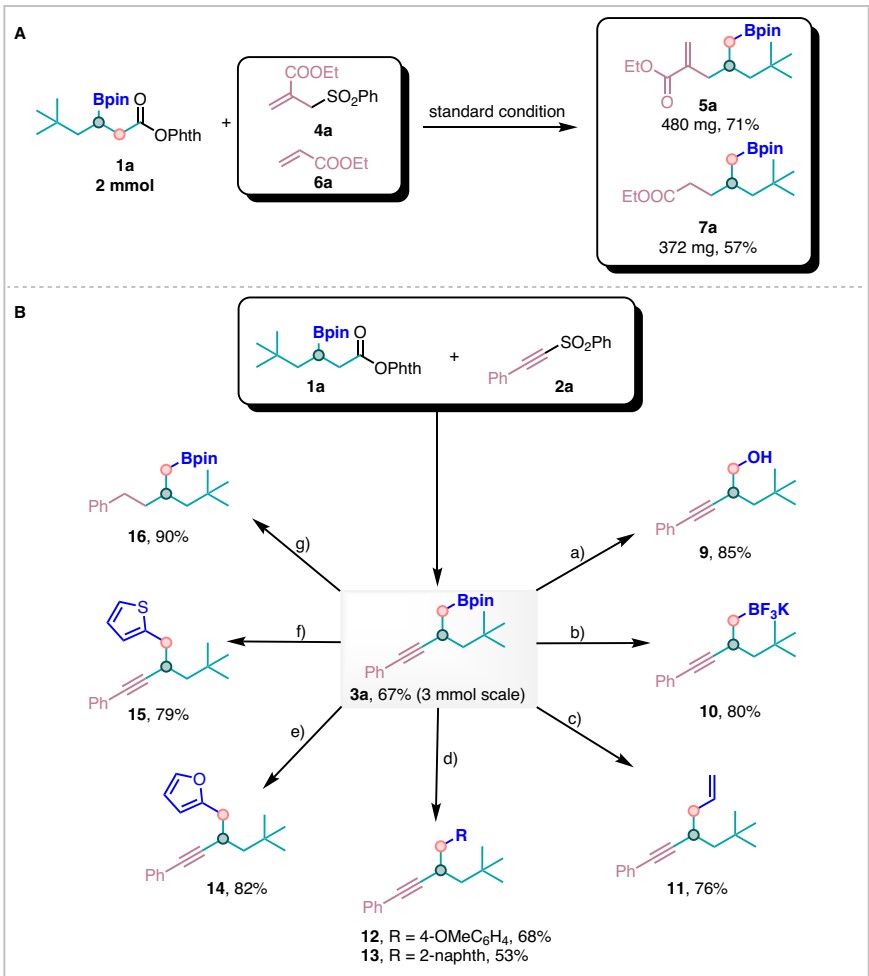

**Fig. 6 | Synthetic applications.** Reaction conditions: **a** NaBO₃·H₂O, THF/H₂O; **b** KHF₂, MeOH; **c** vinylmagnesium bromide, THF, I₂, MeONa; **d** Pd(OAc)₂, BINAP, NaOH, THF/H₂O; **e** n-BuLi, furan, compound **3a**, NBS; **f** n-BuLi, thiophene, compound **3a**; **g** Pd/C, H₂, THF.

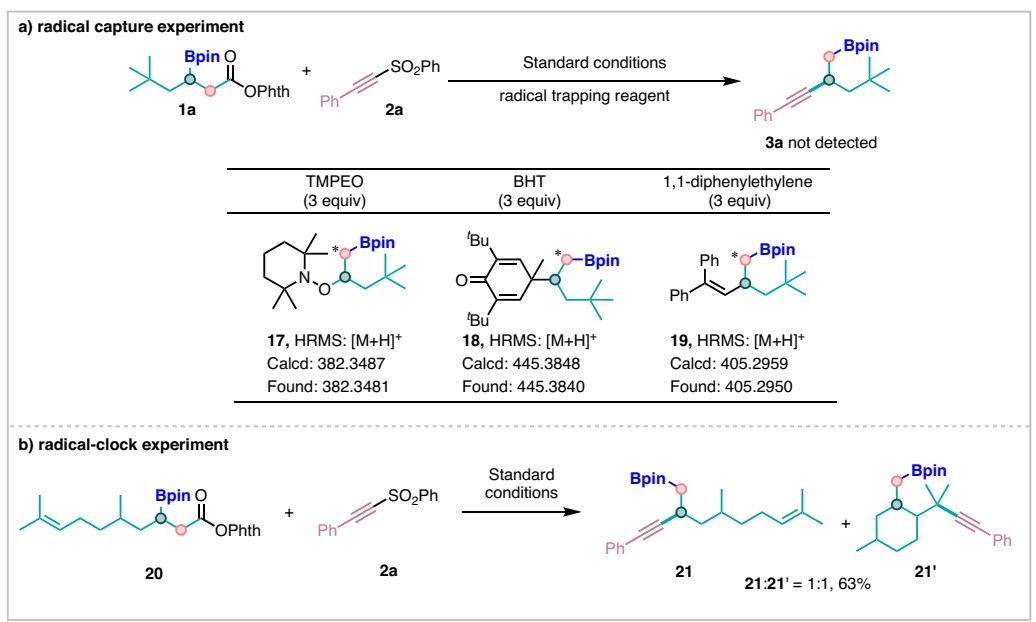

**Fig. 7 | Mechanistic experiments. a** Radical capture experiment. **b** Radical-clock experiment.

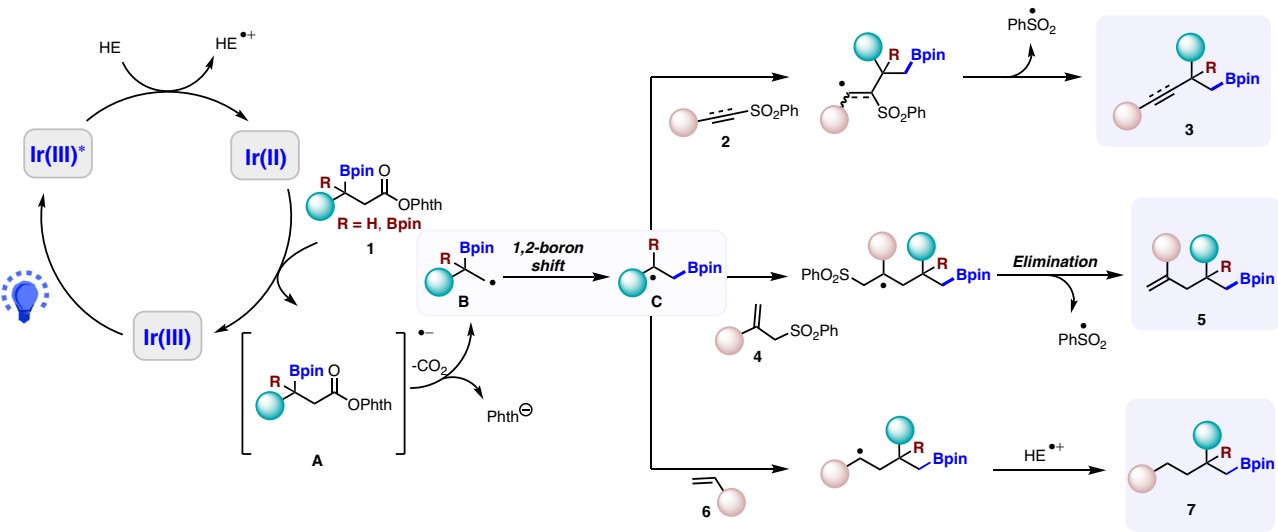

**Fig. 8 | Proposed mechanism.** β-boryl NHPI ester's decarboxylation under the action of light would produce β-boryl alkyl radicals, which undergo a radical 1,2-boron shift to give a translocated C-radical for further capture of versatile radical acceptors.

## Data availability

The data that support the findings of this study are available within the article and its Supplementary Information files. All other data are available from the corresponding author upon request. The X-ray crystallographic coordinates for structures **1a** and **3e** reported in this article have been deposited at the Cambridge Crystallographic Data Centre (CCDC), under deposition numbers CCDC 2247689 (**1a**) and 2242317 (**3e**). These data can be obtained free of charge from The Cambridge Crystallographic Data Centre via http://www.ccdc.cam.ac.uk/data_request/cif.

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

## Acknowledgements

Financial support from National Natural Science Foundation of China (21602065 to J.X., 21931013 and 22271105 to Q.S.), the Natural Science Foundation of Fujian Province (2022J02009 to Q.S.), Promotion Program for Young and Middle-aged Teacher in Science and Technology Research of Huaqiao University (ZQN-705 to J.X.) and Open Research Fund of School of Chemistry and Chemical Engineering, Henan Normal University (to Q.S.) is gratefully acknowledged. We also thank the Instrumental Analysis Center of Huaqiao University for its analysis support.

## Author contributions

Q.S. and J.X. conceived and directed the project. Y.G., X.W., C.L., and J.S. performed experiments. J.X. and Y.G. prepared the Supplementary Information. Q.S. and Y.G. prepared the manuscript. All authors discussed the results and commented on the manuscript.

## Competing interests

The authors declare no competing interests.
