## [Peer Review File · Nature Communications]

Decarboxylation of Beta-Boryl NHPI Esters Enables Radical 1,2-Boron Shift for the Assembly of Versatile OrganoboronsReviewers' Comments:

Reviewer #1:

Remarks to the Author:

The manuscript submitted by Song and coworkers reports visible-light-induced photoredox neutral decarboxylative radical cross-coupling reactions. This reaction undergoes a radical 1,2-boron shift to generate translocated C-radicals, which are subsequently captured by versatile radical acceptors, leading to the formation of valuable mono-organoborons or 1,2-bis(boronic) esters. This methodology demonstrates broad functional group tolerance, and a range of diverse beta-boryl and beta-gem-bis(boryl) NHPI esters, as well as versatile radical acceptors, can be applied to these migratory coupling reactions. Substrates containing sensitive functional groups or those derived from natural products, show excellent reactivity and high compatibility, further confirming the robustness and efficiency of this methodology. Additionally, the application of the products allows for the synthesis of versatile molecules through simple modifications. These demonstrate the products could serve as potential building blocks in organic synthesis. The authors run radical clock and capture experiments to elucidate the reaction mechanism, and propose a rational mechanism based on the results of these experiments. The manuscript is logically structured, and the references are appropriately cited. Therefore, this work is suitable for publication in Nature Communications with minor revisions.

1) The authors have developed an efficient method to afford the migratory coupling products at the more substituted sites. However, were any products detected at the less substituted site of beta-boryl and beta-gem-bis(boryl) NHPI esters without 1,2-boryl migration? Considering these reactions initially generate primary free radicals through the decarboxylation of NHPI esters, these intermediates should be capable of being captured theoretically.

2) The beta-boryl NHPI esters (1u, 1v and 1aa in Figure 3) possess alkenyl moieties. Did the author observe any byproducts resulting from intramolecular radical cyclization?

3) The author demonstrated that beta-gem-bis(boryl) NHPI esters are also suitable precursors for these migratory coupling reactions. While I understand that boron-stabilized free radicals are typically stable and often display low activity, achieving this reaction appears to be quite challenging. I speculate that beta-gem-bis(boryl) NHPI esters could potentially be extended to synergistic catalysis, and perhaps even to asymmetric catalysis, which extends beyond the scope of the present study.

4) Three different radical scavengers, namely TEMPO, BHT and 1,1-diphenylethylene, were used in these reactions, and the desired radical trapping products could be detected by HRMS. However, the other capture product of primary radical intermediates shares the same molecular weight as those of the secondary radical from 1,2-boron shift. If these intermediates could be isolated and confirmed through NMR analysis, the mechanism of the radical 1,2-boryl migration would be more effectively rationalized.

5) Since the review (Eur. J. Org. Chem. 2022, e202101463) is relevant to the topic about 1,2-boron shift, please cite this in the references.

6) There are some typos that need to be addressed.

Page 3: Please change "the resulted adduct radical" to "the resulted radical adduct"

Page 6, Please change "various aliphatic 1,2-diboron" to "various aliphatic 1,2-diborons"

Page 7: There is a space between "Ir(ppy)₃" and "was"

Page 9: The following sentence is unclear and should be revised: "For example, a series of natural products (Dodecyl aldehyde, Cyclamen aldehyde, Lily aldehyde) were also successfully introduced into the corresponding final products (3p, 3q and 3r) without loss of efficiency."

Page 14: Please change "R = 4-OMePh" to "R = 4-OMeC₆H₄" in Figure 6.

Page 15. "Next, carbon homologations through the incorporation of an alkene group" needs to be changed to "Next, carbon homologations were achieved through the incorporation of an alkene group"

Reviewer #2:

Remarks to the Author:

The manuscript submitted by Xu, Song and coworkers designed and synthesized boron-containing NHPI esters as radical precursors. The decarboxylation of such boron compounds under photocatalysis conditions would produce β -boryl alkyl radicals, which could be used for the radical 1,2-boron migration to give a translocated C-radical for further capture of a range of radical acceptors. An easily modified boronate group enhances the value of this approach. The synthetic utility of the reaction was demonstrated with a series of valued commercial chemicals and pharmaceutically interesting molecules. The mild reaction conditions and easy operations enable this reaction to be a practical strategy and gram-scale synthesis further demonstrates the synthetic versatility. The reaction is tolerable to a wide range of functional groups, making it an alternative strategy for the construction of primary organoborons via a novel radical 1,2-boron shift mode. The synthetic transformations further demonstrate the practical values of this strategy.

The supporting information is well prepared and detailed. All the compounds were well characterized. The quality and originality of the science is outstanding and the chemistry herein represents a breakthrough in the fields of both 1,2-boron shift and the synthesis of versatile primary boronates. The novelty of the chemistry, technical advances, and the insights provided by the detailed mechanistic study make the manuscript valuable and suitable for publication in Nature Communications.

Some minor technical issues as detailed below should be addressed before final acceptance:

1. In Figure 5, the authors state that the reaction works well for electron-deficient alkenes, did the authors try electron-neutral alkenes and styrene as radical acceptors?
2. A very large number of examples were demonstrated, however, most of them are restricted to variations of primary and secondary alkyl groups in β -boryl NHPI esters, more challenging substrates, such as tertiary alkyl groups were not present. I consider that at least one more example of such substrate would be of great value.
3. In Figure 6, it may be more intuitive to abbreviate the reaction conditions at each step. It would be better to hide the reagent equivalents, reaction temperature and time. A full description of the reaction conditions in the supporting information is sufficient.
4. Did the authors consider the use of other boron containing groups, such as B₂neop₂?
5. In addition to alkynyl aryl sulfones, alkenyl aryl sulfones, allyl sulfone and electron-deficient alkenes, other aryl sulfone compounds, such as TsN₃, TsCN, are compatible with this reaction?

Point-by-point response to Reviewer(s)' Comments

REVIEWER COMMENTS

Reviewer #1 (Remarks to the Author):

The manuscript submitted by Song and coworkers reports visible-light-induced photoredox neutral decarboxylative radical cross-coupling reactions. This reaction undergoes a radical 1,2-boron shift to generate translocated C-radicals, which are subsequently captured by versatile radical acceptors, leading to the formation of valuable mono-organoborons or 1,2-bis(boronic) esters. This methodology demonstrates broad functional group tolerance, and a range of diverse beta-boryl and beta-gem-bis(boryl) NHPI esters, as well as versatile radical acceptors, can be applied to these migratory coupling reactions. Substrates containing sensitive functional groups or those derived from natural products, show excellent reactivity and high compatibility, further confirming the robustness and efficiency of this methodology. Additionally, the application of the products allows for the synthesis of versatile molecules through simple modifications. These demonstrate the products could serve as potential building blocks in organic synthesis. The authors run radical clock and capture experiments to elucidate the reaction mechanism, and propose a rational mechanism based on the results of these experiments. The manuscript is logically structured, and the references are appropriately cited. Therefore, this work is suitable for publication in *Nature Communications* with minor revisions.

Response: We sincerely thank this reviewer for the favorable comments on our work. We really appreciate it.

1) The authors have developed an efficient method to afford the migratory coupling products at the more substituted sites. However, were any products detected at the less substituted site of beta-boryl and beta-gem-bis(boryl) NHPI esters without 1,2-boryl migration? Considering these reactions initially generate primary free radicals through the decarboxylation of NHPI esters, these intermediates should be capable of being captured theoretically.

Response: We thank this reviewer for the constructive comments. During our condition screening with 1,3-dioxoisindolin-2-yl 4,4-dimethyl-3-(4,4,5,5-tetramethyl-1,3,2-dioxaborolan-2-yl)pentanoate (**1a**) and ((phenylethynyl)sulfonyl)benzene (**2a**) as substrates, no byproducts were found without 1,2-boryl migration. However, when we used different beta-boryl NHPI esters for substrate expansion, we indeed found that such byproducts were formed without 1,2-boryl migration for allyl sulfones (see below). Due to the extremely similar polarity between the migratory coupling products and the byproducts without 1,2-boryl migration, we could only obtain a mixture combining the two isomers instead of obtaining the pure products without 1,2-boryl migration. Through nuclear magnetic resonance, we found that the ratio of the two products is ca. 1:1.

2) The beta-boryl NHPI esters (**1u**, **1v** and **1aa** in Figure 3) possess alkenyl moieties. Did the author observe any byproducts resulting from intramolecular radical cyclization?

Response: We sincerely thank this reviewer for the inspiring comments. For **1u**, **1v** and **1aa** in Figure 3, we didn't detect any byproducts resulting from intramolecular radical cyclization. We speculated that the reaction rate between free radicals and alkenyl arylsulfones may be faster than the rate of intramolecular cyclization in this condition, so we only obtained the migratory coupling products.

3) The author demonstrated that beta-gem-bis(boryl) NHPI esters are also suitable precursors for these migratory coupling reactions. While I understand that boron-stabilized free radicals are typically stable and often display low activity, achieving this reaction appears to be quite challenging. I speculate that beta-gem-bis(boryl) NHPI esters could potentially be extended to synergistic catalysis, and perhaps even to asymmetric catalysis, which extends beyond the scope of the present study.

Response: We deeply thank this reviewer for the constructive and inspiring suggestions. In fact, we are currently attempting to construct chiral centers using transition metal catalysis with this transformation.

4) Three different radical scavengers, namely TEMPO, BHT and 1,1-diphenylethylene,

were used in these reactions, and the desired radical trapping products could be detected by HRMS. However, the other capture product of primary radical intermediates shares the same molecular weight as those of the secondary radical from 1,2-boron shift. If these intermediates could be isolated and confirmed through NMR analysis, the mechanism of the radical 1,2-boryl migration would be more effectively rationalized.

Response: We deeply thank this reviewer for raising this issue. When radical scavengers (TEMPO, BHT and 1,1-diphenylethylene) were added under standard conditions, the reaction system became chaotic. We attempted to isolate and obtain the radical trapping products, but the obtained radical trapping products were quite messy through NMR analysis. Therefore, the captured products can only be detected through HMRS.

5) Since the review (Eur. J. Org. Chem. 2022, e202101463) is relevant to the topic about 1,2-boron shift, please cite this in the references.

Response: We thank this reviewer for pointing it out, we have cited the article as ref. 42. Please see our revised manuscript.

6) There are some typos that need to be addressed.

Page 3: Please change “the resulted adduct radical” to “the resulted radical adduct”

Response: We sincerely thank this reviewer for pointing it out. Sorry for the negligence. We have changed the word in the original manuscript. Please see our revised manuscript.

Page 6, Please change “various aliphatic 1,2-diboron” to “various aliphatic 1,2-diborons”

Response: We sincerely thank this reviewer for pointing it out, we have corrected this mistake. Please see our revised manuscript.

Page 7: There is a space between “Ir(ppy)₃” and “was”

Response: Sorry for the negligence. We have corrected this mistake. Please see our revised manuscript.

Page 9: The following sentence is unclear and should be revised: “For example, a series of natural products (Dodecyl aldehyde, Cyclamen aldehyde, Lily aldehyde) were also successfully introduced into the corresponding final products (3p, 3q and 3r) without loss of efficiency.”

Response: We thank this reviewer for pointing it out, we have revised this sentence. Please see our revised manuscript.

Page 14: Please change “R = 4-OMePh” to “R = 4-OMeC₆H₄” in Figure 6.

Response: Thanks for pointing it out, we have corrected this mistake. Please see our revised manuscript.

Page 15. “Next, carbon homologations through the incorporation of an alkene group” needs to be changed to “Next, carbon homologations were achieved through the

incorporation of an alkene group”

Response: Thanks for pointing it out, we have corrected this mistake. Please see our revised manuscript.

We deeply thank this reviewer for his/her constructive and inspiring suggestions, which greatly improve our manuscript, we really appreciate it.

Reviewer #2 (Remarks to the Author):

The manuscript submitted by Xu, Song and coworkers designed and synthesized boron-containing NHPI esters as radical precursors. The decarboxylation of such boron compounds under photocatalysis conditions would produce β -boryl alkyl radicals, which could be used for the radical 1,2-boron migration to give a translocated C-radical for further capture of a range of radical acceptors. An easily modified boronate group enhances the value of this approach. The synthetic utility of the reaction was demonstrated with a series of valued commercial chemicals and pharmaceutically interesting molecules. The mild reaction conditions and easy operations enable this reaction to be a practical strategy and gram-scale synthesis further demonstrates the synthetic versatility. The reaction is tolerable to a wide range of functional groups, making it an alternative strategy for the construction of primary organoborons via a novel radical 1,2-boron shift mode. The synthetic transformations further demonstrate the practical values of this strategy.

The supporting information is well prepared and detailed. All the compounds were well characterized.

The quality and originality of the science is outstanding and the chemistry herein represents a breakthrough in the fields of both 1,2-boron shift and the synthesis of versatile primary boronates. The novelty of the chemistry, technical advances, and the insights provided by the detailed mechanistic study make the manuscript valuable and suitable for publication in *Nature Communications*.

Response: We sincerely thank this reviewer for the favorable comments on our entitled manuscript, we really appreciate it.

Some minor technical issues as detailed below should be addressed before final acceptance:

1. In Figure 5, the authors state that the reaction works well for electron-deficient alkenes, did the authors try electron-neutral alkenes and styrene as radical acceptors?

Response: We thank this reviewer for the constructive comments. We tried several other radical acceptors, such as styrene, unactivated olefins, which are not compatible with this reaction system.

2. A very large number of examples were demonstrated, however, most of them are restricted to variations of primary and secondary alkyl groups in β -boryl NHPI esters, more challenging substrates, such as tertiary alkyl groups were not present. I consider that at least one more example of such substrate would be of great value.

Response: We deeply thank this reviewer for the constructive comments and

suggestions. Per the request, we prepared a tertiary alkyl group (cyclobutyl) in β -boryl NHPI ester and subject it to our standard conditions, to our delight, the corresponding desired product **3p** was obtained albeit in lower yield (33%). We added it to Figure 2, please see our revised manuscript.

3. In Figure 6, it may be more intuitive to abbreviate the reaction conditions at each step. It would be better to hide the reagent equivalents, reaction temperature and time. A full description of the reaction conditions in the supporting information is sufficient.

Response: We sincerely thank this reviewer for pointing it out, per the request, we have hidden the reagent equivalents, reaction temperature and time. Please see our revised manuscript.

4. Did the authors consider the use of other boron containing groups, such as B₂neop₂?

Response: We deeply thank this reviewer for pointing it out, we have examined β-boryl NHPI esters (using B₂neop₂ as the boron source for 1,2-boron shift). The substrate is also compatible with our reaction system. Please see our revised manuscript and supporting information.

5. In addition to alkynyl aryl sulfones, alkenyl aryl sulfones, allyl sulfone and electron-deficient alkenes, other aryl sulfone compounds, such as TsN₃, TsCN, are compatible with this reaction?

Response: We thank this reviewer for pointing it out. Actually during substrate scope exploration, we also considered whether other aryl sulfone compounds compatible with this reaction system and we did many trials, however it turned out that TsN₃ and TsCN are not compatible with our system.

Reviewers' Comments:

Reviewer #1:

Remarks to the Author:

Prof. Song and coworkers have made a considerable effort to address my concerns. I recommend the manuscript to be published in Nat. Commun. without further scientific changes.

Reviewer #2:

Remarks to the Author:

In the revised version of the manuscript, Xu, Song, and their colleagues successfully addressed all the shortcomings of the original submission (in my opinion). This article is appropriate for publication at this stage.

Point-by-Point Response to the reviewers' comments

REVIEWER COMMENTS

Reviewer #1:

Comments for the Author:

Prof. Song and coworkers have made a considerable effort to address my concerns. I recommend the manuscript to be published in Nat. Commun. without further scientific changes.

Our response: We sincerely thank this reviewer for his/her favorable comments on our manuscript, meanwhile, we also would like to acknowledge this reviewer for his/her contribution to improve the quality of our manuscript.

Reviewer #2:

Comments for the Author:

In the revised version of the manuscript, Xu, Song, and their colleagues successfully addressed all the shortcomings of the original submission (in my opinion). This article is appropriate for publication at this stage.

Our response: We sincerely thank this reviewer for his/her favorable comments on our manuscript, meanwhile, we also would like to acknowledge this reviewer for his/her contribution to improving the quality of our manuscript.